# Design and Preparation of Sensing Surfaces for Capacitive Biodetection

**DOI:** 10.3390/bios13010017

**Published:** 2022-12-23

**Authors:** Perrine Robin, Sandrine Gerber-Lemaire

**Affiliations:** Group for Functionalized Biomaterials, Institute of Chemical Sciences and Engineering, Ecole Polytechnique Fédérale de Lausanne, 1015 Lausanne, Switzerland

**Keywords:** non-faradaic impedance, capacitive detection, biosensors, electrochemical impedance spectroscopy, surface functionalization, biomolecules immobilization, interdigitated electrodes

## Abstract

Despite their high sensitivity and their suitability for miniaturization, biosensors are still limited for clinical applications due to the lack of reproducibility and specificity of their detection performance. The design and preparation of sensing surfaces are suspected to be a cause of these limitations. Here, we first present an updated overview of the current state of use of capacitive biosensors in a medical context. Then, we summarize the encountered strategies for the fabrication of capacitive biosensing surfaces. Finally, we describe the characteristics which govern the performance of the sensing surfaces, along with recent developments that were suggested to overcome their main current limitations.

## 1. Introduction

Affinity-based biosensors operate by specifically capturing a biological target with biological or synthetic capture agents such as aptamers, DNAzymes, single stranded DNAs or antibodies. The binding event between the capture molecule and its target is later translated into a readable signal. 

Labelling the target molecules with fluorophores, magnetic beads, quantum dots or enzymes, was reported to facilitate and amplify the readout signal. However, labelled-detection methods are expensive and necessitate multi-step processes, hence are limited for real-time detection. Label-free methods are therefore of interest for high throughput biomolecules screening, portable devices and suitable for large-scale production [1]. 

A variety of transducing methods have been described to translate the binding event between the targeted molecules and the probes (Figure 1). Conventional approaches include optical, electrical or mass-sensitive techniques [2]. Electrical biosensors are great candidates for miniaturized, portable and label-free detection, relying on potentiometric, voltammetric, amperometric or impedimetric readout signals. Impedance-based sensors are themselves divided in faradaic or non-faradaic detection. Faradaic biosensors refer to the detection of charge transfer across a membrane [3]. They often rely on electrochemical impedance spectroscopy (EIS) [4] which detects the binding events via the change of electron transfer resistance and double layer capacitance within a frequency range [5]. However, faradaic detection is complex as it necessitates a wide window of frequencies [4]. It also requires the addition of potentially hazardous redox couples, that can degrade biomolecules [4,6]. On the other hand, non-faradaic based sensors, also called capacitive sensors or third-generation biosensors [7], detect the changes of capacitance at the electrode surface caused by the molecular binding events. These sensors have a high-sensitivity potential [4], and do not require the addition of external reagent unlike other conventional methods, such as in situ hybridization or enzyme linked immunosorbent assays (ELISA) [4,8]. They offer a simple and rapid detection that can be inserted into portable devices [9]. Additionally, non-faradaic capacitive detection does not require trained laboratory personal or samples preparation, and is therefore interesting for point-of-care applications [4,10,11,12]. Unfortunately, it has been reported that capacitive biodetection suffers from poor to poor reproducibility [4,12,13,14,15] and large standard deviation [16,17], preventing their translation for clinical application. These limitations have been suspected to arise form sensing surface parameters such as its cleanliness [4,18], homogeneity [4] and insulation [10,19,20,21]. 

Previous review articles focused on the different types of capacitive sensors [3,22,23], geometries of electrodes [3], the use of nanomaterials for signal enhancement [5], the preparation of the electrodes by molecular imprinting [1,24]. A specific insight into capacitive immunosensors was also disclosed [25]. Here, we present the importance of the preparation of sensing surfaces to overcome the limitations which capacitive biosensors face in clinical applications. Towards this goal, we will discuss the different functionalization strategies to immobilize the capture molecules on the electrodes, in addition to the influence of surface properties (cleanliness and homogeneity) on the performance and reliability of the resulting biosensors. Additionally, methods to obtain well-insulated surfaces and efforts made to avoid non-specific binding of the target molecules at the surface will be examined. 

## 2. Methodologies for Capacitive Biosensor Detection

Two main electrode geometries were reported for capacitive sensing, leading to two distinct capacitive methodologies for the detection of binding events between capture molecules and their targets. The most common is based on potentiostatic capacitance measured at the electrode/solution interface. In this case, the capture molecules are immobilized on the working electrodes. As an alternative, interdigitated electrodes have received growing attention over the last three decades for capacitive detection [3]. In this case, the recognition elements are immobilized on a substrate in between the electrodes, which undergo capacitance changes upon binding to the molecular targets [22]. 

### 2.1. Potentiostatic Capacitance

Potentiostatic capacitance measurements at the electrode/solution interface is based on the theory of the electrical double-layer. The electrodes are generally made of a conductive metal, often covered by an insulating layer on which the capture biomolecules are immobilized. When the electrode is immersed in an electrolyte solution, at a given potential, the surface charge (q_m_) and solution charge (q_s_) equilibrate according to q_m_ = −q_s_. Charged species and dipoles contained in the solution will orientate towards the surface, forming the double-layer. It was demonstrated experimentally that the system can be represented as three capacitors in series, as illustrated in Figure 2. The insulating layer represents the first capacitor, *C_ins_*. The second capacitor, *C_rec_*, corresponds to the immobilized molecules and their target, in addition to the double layer. The last capacitor (*C_d_*) represents the diffuse layer. Overall, the capacitance of the system *C_tot_* is given by Equation (1) [22].
(1)$$C_{tot}=\frac{1}{\frac{1}{C_{ins}}+\frac{1}{C_{rec}}+\frac{1}{C_{d}}}$$

When the target molecule binds to the capture probe, it results in a change of *C_rec_* that can be detected via the monitoring of the total capacitance, *C_tot_*. As shown in the Equation (1), *C_tot_* is dominated by the smallest capacitance of the three. Therefore, if present, the insulating layer should be thin with a high dielectric constant to result in a high *C_ins_* value. 

### 2.2. Interdigitated Electrodes

Capacitive sensors based on interdigitated electrodes (IDE) are also called interdigital capacitive sensors. The use of these electrodes results in enhanced impedance changes, higher signal-to-noise ratio, and increased speed of detection [4].

The geometry of IDEs is illustrated in Figure 3. Usually, IDEs are fabricated by lithography techniques on glass substrates or silicon wafers. The electrodes width and spacing can vary from tenths of nanometers to tenths of microns [3]. In this configuration, the sensor surface consists of two parallel metal electrodes separated by a dielectric surface. The capture probe can be immobilized on the electrodes, on the insulating surface, in between, or on both [19]. The capacitance of the electrodes is linked to the dielectric constant of the surface in between the electrodes with the Equation (2) [22].
(2)$$C=\frac{\varepsilon \times \varepsilon_{0}\times A}{d}$$
where ε stands for the dielectric constant of the substrate between the plates, ε_0_ = 0.8.85419 pF/m (vacuum permittivity), *A* is the area of the electrodes and *d* the distance between them.

Thus, when molecule binds to the surface, it causes a change in the dielectric properties of the substrate, that can be retrieved by monitoring the changes of capacitance of the electrodes. 

Instead of measuring a capacitance change, an alternative approach consists of measuring the changes of distances between the capacitor plates. These devices are composed of a rigid electrode and a flexible one, often a membrane. Capacitive membranes were previously reported by Tsouti et al. [3] and therefore are not further described. 

## 3. Current State of Capacitive Biosensors Used in Clinical Applications

Due to the variety of capture probes which can be immobilized on the electrodes surface, capacitive biosensors can be designed for a wide range of medical applications. Viruses, unicellular and disease markers were detected in biological fluids, via non-faradaic measurements. The following sections focus on the most recent studies which enabled to significantly decrease the limits of detection (LoD) compared to previous systems and/or addressed capacitive detection in complex biological samples. 

### 3.1. Infections

Immunosensors have been reported for the capacitive detection of viruses such as Influenza virus [26,27], foot and mouth disease [28], Hepatitis B [29], Norovirus [30], Zika virus [31] and SARS-CoV-2 [32]. However, genosensors were more widely described for viral capacitive detection, and generally displayed lower LoD than immunosensors. In 1999, Bergreen et al. could detect Cytomegalovirus with a limit of detection of 0.2 aM from DNA standard fragments [13]. More recently, commercial DNA sequence of West Nile virus could be detected with a limit of detection (LoD) of 1.5 aM [4]. 

While these studies investigated the specificity of the obtained sensors with non-targeted genes, none of them reported the detection of viral targets in complex biological samples. Matrix effect, coming from the interaction of non-targeted biomolecules with the sensors surface, can have deleterious effect on the sensing efficiency. For example, in 2017, Cheng et al. reported a genosensor for the detection of Herpes 1 virus [33]. While a LoD of 10.7 aM could be achieved in standard buffer, the limit increased to 0.21 fM in neat serum, likely due to unwanted interactions of non-targeted biomolecules with the sensor surface [33]. To our knowledge, no capacitive sensors displaying attomolar detection has been reported to detect viruses in real biological samples.

Diagnosing a viral infection can also be performed by recognizing antibodies, i.e., performing serological tests. In that regard, Zeng et al. recently reported an impedimetric biosensor that can detect SARS-CoV-2 antibodies (Abs). The authors studied two enhancement techniques (illustrated in Figure 4) to improve the LoD of the system, (1) by probing the targeted Abs with a gold nanoparticle (AuNP)-tagged secondary antibody and (2) by using dielectric electrophoresis (DEP). In the case of the AuNPs enhancement strategy, the presence of the nanoparticles enhances the measurable signal, that can reduce the LoD value. In the case of DEP enhancement, targeted biomolecules can be selectively moved and concentrated to the sensing surface due to the dielectric properties of each molecule. While a LoD of 2 μg/mL was obtained with the DEP enhancement technique, the authors could detect down to 200 ng/mL with AuNPs [34]. 

Non-faradaic impedimetric sensors were also reported for other type of pathogens, including protozoan, worms and bacteria. In 2022, Figueroa-Miranda et al. reported a aptasensor based on a graphene surface for the detection of a malaria marker, *Plasmodium falciparum* lactate dehydrogenase, with a limit of detection of 0.78 fM in diluted human serum [35], allowing the detection of low-density parasitemia, and surpassing the LoD previously achieved via Faradaic detection [36,37,38].

Worm’s antigens were also detected via non-faradaic impedance measurements. Zhou et al., reported the use of gold electrodes modified with *Schistosoma japonicum* antibodies to detect the worms’ antigens. A LoD of 0.1 ng/mL was reached in PBS., and the selectivity of the sensor was assessed by comparing the capacitance changes with other proteins [39].

The presence of bacteria can either be detected via the presence of toxins, or directly detecting the unicellular organisms. While capacitive detection of toxins was reported in food [40,41] or in water [42], detection of toxins has not been reported for diagnosis purposes. On the contrary, direct detection of bacteria has been reported and is described in the Unicellular detection section.

### 3.2. Cancer, Chronic and Inflammatory Diseases

Early detection and diagnosis of cancer, chronic or inflammatory diseases can drastically improve the chances of survival [43,44,45]. Capacitive detection of such biomarkers has been used in this sense for a variety of pathologies. 

A variety of cancer biomarkers were selected as targeted molecules for early cancer detection. Recent studies pointed to the detection of markers with a sufficient LoD to enable cancer early diagnosis. For example, in 2018, Arya et al. [46] reported a LoD of 0.1 ng/mL in non-diluted serum. As patients suffering from breast cancer possess around 14 to 75 ng/mL of Her2 markers in their blood, this aptasensor could be used in the future for diagnosis [46]. 

The detection of marker for chronic and inflammatory diseases was also reported via capacitance measurements. For example, the capacitive detection of C-reactive protein (CRP), a biomarker for cardiovascular disease risks, sepsis and other tissue inflammation was extensively studied [11,12,47,48,49]. A decade later, Macwan et al. [48], could surpass the sensitivity of previous CRP electrochemical sensors [11,12] and reached a 10 fg/mL LoD in both PBS buffer and serum by switching to an interdigitated sensing device integrated sputtered with nanofibers to enhance the sensitivity and selectivity of the sensing surface. Human chitinase-3-like protein 1, a marker for tissue inflammation and cardiac disease was also detected by capacitive systems with a LoD of 0.07 μg/L. This is 300 times lower than ELISA method, currently used for this marker detection [50]. However, this study was performed in diluted serum to prevent the matrix effect and complementary studies should thus be performed to assess the LoD in real biological sample. Other biomarkers have been targeted for chronic-disease diagnosis, such as LDL-cholesterol [6], interleukin-3 [51] or transferrin [20,52]. Nampt, an obesity marker for which Park et al. improved the binding affinity with the surface compared to previously reported techniques, could be detected in the nanomolar range [53]. Finally, Kumar Sharam et al. recently reported the detection of amyloid beta within 5 s with a LoD of 0.1 fg/mL for the diagnostic of Alzheimer disease [54]. 

### 3.3. Unicellular Organisms

Capacitive sensors for whole cell detection and analysis were developed over the last decade. The first occurrence of unicellular organism detection by non-faradaic measurements dates back 2004, for *E. Coli* detection in food samples by using an immunosensor targeting antigens present at the bacteria surface [55]. Since then, the detection of *Salmonella* (bacteria), *Cryptosporidium* (protozoan) in food or water samples was disclosed. In 2013, Couniot et al. also reported a simulation for optimal design of capacitive sensors for bacteria detection [56]. Recently, Borsel-Oliu et al. reported the use of a 3D IDE platform that evaluates the response of bacteria to antibiotics [57]. This platform could in the future be used for a variety of toxicity evaluations. 

Non-faradaic biosensors have also been developed for eukaryotic cell detection. In 2022, Zhang et al. [58], detailed the detection of peripheral blood mononuclear cell (PBMCs), that can indicate the immune function state of a patient. While PMBCs cells are normally found in concentrations ranging from 0.7 to 6.2·10^6^ cells/mL, their sensor displayed a LoD of 10^4^ PBMCs/mL, with the possibility to quantify the cells. However, further experiments would be needed to assess the LoD in real samples for potential clinical applications.

Finally, yeast detection has also been reported with a non-pathogenic strain, *Saccharomyces cerevisiae*. A LoD value of 0.1 ng/mL was achieved, with a detection range of 0.4 to 18 ng/mL [59]. We expect that further developments could lead to the detection of pathogenic strains for clinical use.

Viruses, unicellular organisms and diseases markers detected via capacitive biosensing are summarized in Table 1.

## 4. Capacitive Sensing Surfaces Preparation, and Limitations

### 4.1. Capacitive Sensors Limitations

Capacitance biosensors, whether they are based on IDEs or potentiostatic capacitance measurements, necessitate the immobilization of biomolecules on a surface. In the case of IDEs, the molecules are immobilized on the interface between the electrodes. For potentiostatic capacitance measurements, probes are immobilized directly on the electrode, eventually covered with an insulating layer (see in Figure 2 and Figure 3). The capture molecules can be antibodies, to further detect an antigen or a pathogen (immunosensor), single-strand DNAs, for the detection of RNA or DNA single strands (genosensor) or aptamers, for the fabrication of aptasensors.

Unfortunately, even after years of progress in capacitive biodetection, challenges remain. Poor reproducibility [4,12,12,13,14,15] and large standard-deviation [16,17] of capacitive biosensors have been reported and linked to non-optimal parameters of their sensing surfaces [4,13,19]. Critical parameters in the capacitive sensing surfaces preparation have been raised, such as the probe immobilization strategy [13,19,52], surface cleanliness [4,18], homogeneity [4] and insulation [10,19,20,21]. Obtaining a high specificity with capacitive sensors is also challenging as any adsorbed biomolecules at the sensing surface can prevent target binding or generate a false-positive signal [10,16,17,62]. After describing the most common strategies for the preparation of sensing surfaces, the surface parameters suspected to affect capacitive sensors behavior are presented, along with the solutions recently reported to overcome these limitations. 

### 4.2. Biomolecules Immobilization Techniques

Various immobilization strategies have been reported for the immobilization of capture molecules on capacitive sensing surfaces. The functionalization procedures may depend on the type of electrodes used and the nature of capture molecules immobilized. Selection of the functionalization pathway for a capacitive sensor is crucial, as it can affect drastically its performance [23].

While electrodes are generally made from metals such as gold [4,8,11,12,14,19,21,26,27,28,29,33,39,40,46,47,48,50,53,55,58,62,70,71,72,73,74,75,76,77,78,79], platinum [41], graphene [35,80], glass carbon [7,52] or aluminum [33,64,68], electrodes made of titanium [9,60,63,81,82], nickel [66], or silicium [65,83,84,85] are less described even though they are great candidates for capacitive measurements. Indeed, they display high insulation properties, can present smooth surfaces with homogeneous functionalities for capture probe immobilization [23]. Finally, few more exotic materials have been reported as electrodes material. For example, tantalum was selected for antibodies [86] or bacteria detection [57]. Recently, polymers were reported for the production of electrodes for capacitive biodetection. Park et al. described the use of electrodes made of a conductive polymer layer of PEDOT:PSS for the detection of SARS-CoV-2 [32]. Frias et al. used polyvinyl alcohol, alginate and polyaniline to fabricate electrodes for Zika virus detection [31]. 

Gold electrodes are generally favored for capacitive measurements as they are frequently encountered in other detection transduction techniques such as surface plasmon resonance, quartz crystal microbalance or reflection absorption IR spectroscopy, thus allowing for dual-mode detection and direct comparison of readout signals [18].

Two main strategies are commonly described for the immobilization of capture molecules on sensing surfaces. A first method consists of the formation of a self-assembled monolayer (SAM) at the surface of metallic electrodes. This layer can be directly made of the capture molecule, or of a linker that is later used to conjugate the capture molecule [87]. A second methods relies on the deposition a thin insulating film at the surface of the electrodes, generally made of a polymer or silanes, followed by the conjugation of the capture molecule to this first layer [23]. 

The most frequently reported strategies for the immobilization of capture probes on the electrode and/or insulating layer are detailed in the following sections. 

#### 4.2.1. Self-Assembled Monolayer (SAM) Formation on Metal Electrodes 

The adsorption of SAMs of biomolecules on metals and metal oxides is frequently used for the modification of electrodes. Due to the strong affinity of sulfur atoms for gold, the most reported SAM technique is based on the incubation of gold surfaces with thiol-modified biomolecules [87]. 

Thiol-modified single-strand DNA (ssDNA) probes were immobilized on gold electrodes for cytomegalovirus detection [13]. ssDNA immobilization on aluminum electrodes was also performed for the detection of West Nile [4] and Herpes viruses [33], but also on indium oxide for the detection of papillomavirus [61]. The immobilization of thiol-modified aptamers on gold electrodes was described for the detection of obesity markers [53], cancer [46] or inflammatory diseases markers [12], but also thrombin [21], and malaria [62]. Finally, other capture molecules such as affirmers [63], peptide-nucleic acids [74], and epitopes [28], were exploited for diagnosis application.

Another strategy consists of adding a first layer of alkyl-thiol chain on the electrode, to which the capture molecules is further conjugated. This strategy was reported for the immobilization of antibodies [6,11,14,27,48,50,58,69,71,72,75,77,78,79], ssDNA [13] and aptamers [8,80]. Numnuam et al. also reported the immobilization of histones on gold electrodes for the detection of DNA traces [88]. Figure 5 illustrates the differences between direct surface functionalization with thiol-containing biomolecules via SAM formation, and the use of a thiol linker for covalent conjugation of the capture molecule.

The direct immobilization of biomolecules via SAM formation is the simplest way to functionalize metallic surfaces with capture probes [89]. However, it requires the use of thiol-containing biomolecules or the modification of native biomolecules with a thiol group beforehand. Procedures involving a linker were shown to provide more stable functionalized surfaces. Bergreen et al. compared the two immobilization strategies [13]. On one hand, a 26 bases thiol-modified ssDNA was directly immobilized on gold electrodes, while on the other hand, an 8-base probe was coupled via carbodiimide chemistry to a cysteamine monolayer previously deposited on the gold electrode. The pre-functionalization strategy with a linker resulted in enhanced selectivity of the sensor, despite the use a shorter probe. The length of the linker is also of importance. Mirsky et al. studied various SAMs of mercapto-alkyl derivatives deposited on gold electrodes, followed by conjugation with Abs. The study concluded that longer mercapto-alkyl layers were preferable, due to spontaneous desorption of shorter chains. Finally, the use of linkers has also impact on the insulating properties of sensing surfaces which will be discussed in Section 4.5.

#### 4.2.2. Covalent Immobilization on an Insulating Layer

Beside SAM formation on metallic electrodes, capture molecules can also be immobilized on a surface-deposited insulating layer presenting reactive functionalities for further chemical conjugation. 

The most common routes for deposition of the insulating layer include silanization methodologies [19,20,41,60,65,81,82,86] and polymerization strategies. Electropolymerization of tyramine [70,76], phenylenediamine [52], and polyaniline [83,85] were reported for the functionalization of surfaces with amine or carboxylic-containing polymers. Thermal deposition of parylene was also proposed by Jung et al. for the fabrication of a capacitive biosensor [29]. The thickness of the insulating layer is also critical to keep a high sensitivity [29,83]. Using platinum electrodes covered by tin oxide films, Choudhury et al. demonstrated that film thickness below 100 nm resulted in good insulation properties and higher sensitivity than thicker films for the capacitive detection of immunoglobulin [83].

Functionalization of surfaces with polymers instead of silanes was found to improve the sensitivity and specificity of the resulting device, which might be due to surface insulation enhancement [9]. Deposition of a polymeric layer can also be performed with a simpler fabrication procedure than silanization [85]. However, if the sensor is inserted in a microfluidic platform, we believe that the adhesion of the polymer to the surface must be carefully controlled to avoid the removal of the polymer due to the pressure. 

Various chemical linkages can be then used to immobilize capture molecules on the formed insulating layer. Among them, peptide bond coupling has been extensively reported. It has the advantages of being stable under a wide range of pH [90], compatible with many types of surfaces and suitable for the conjugation of a variety of biomolecules due to the presence of amines and/or carboxylic groups in their structure. Carbodiimide chemistry was described for the covalent conjugation of Abs to silanized surfaces [19,86]. The use of cross-linkers was also widely reported. Glutaraldehyde is the most common spacer for the immobilization of biomolecules and was highlighted for the conjugation of Abs to parylene [29], poly(o-phenylenediamine) [52], polytyramine-[70], polyaniline [83,85], and amino-silane modified surfaces [20,81,82]. Similar strategy was reported for the immobilization of phages to a polytyramine insulating layer for the capacitive detection of Salmonella spp. Noticeably, the sensor could be used up to 40 times following alkaline treatment to regenerate the active sensing surface [76]. Other cross-linkers such as N-γ-maleimidobutyryloxy succinimide ester [41,55] or dithiobis (succinimidyl propionate) [47] were disclosed for the conjugation of Abs to capacitive sensing surfaces.

The different methods for covalently conjugating biomolecules to insulating layers are now summarized in Table 2.

As alternative chemical linkage to peptide-bond formation, a few other types of chemical conjugations were reported in the context of the fabrication of capacitive biosensors. When considering nucleic acids as capture molecules, phospharamidite bond formation is of interest for their covalent immobilization. Liao and Cui reported the attachment of a phosphated aptamer with 1-Ethyl-3-(3-dimethylaminopropyl)carbodiimide (EDC) as coupling agent to an APTES-modified surface [65]. Finally, Varlan et al. reported the silanization of TiO_2_ surfaces with glycidoxysilane for the immobilization of antibody via epoxide ring-opening for hormone detection [81]. Figueroa-Miranda et al. described the modification of a graphene oxide surface with pyrene-modified aptamers via π stacking. The resulting sensing surface was later used for malaria detection [35]. EDC/NHS coupling was reported by Yagati et al. for the covalent attachment of aptamers on pyrenebutyric acid previously immobilized on a graphene IDEs via π-π stacking. The obtained aptasensors were studied for the detection of thrombin in blood [80]. The use of biotin/avidin complex formation was also disclosed for the immobilization of antibodies on gold surfaces, addressing the capacitive detection of cardiovascular protein biomarkers [49] and norovirus [30]. 

However, no mention of click chemistry, or use of vinyl sulfone moieties were found in the literature for the conjugation of biomolecules to capacitive sensing surfaces, while encountered in the context of EIS detection [91,92,93].

#### 4.2.3. Influence of the Conjugation Strategy on the Sensor Performance

The immobilization strategy has been highlighted as a crucial parameter to consider for the preparation of a capacitive sensing surface [9,13,19,52].

Castiello et al. recently reported a comparative study of four immobilization techniques for the conjugation of Abs to interdigitated gold electrodes, based on peptide coupling to: (i) a mercapto-alkyl SAM; (ii) an amino-silane monolayer between the electrodes; or a iii) spin-coated poly(methyl methacrylate) (PMMA) layer. These covalent strategies were assessed against passive adsorption of the probe on the electrode (Figure 6). Immobilization to a PMMA spin coated electrode provided the best capacitive behavior due to its smooth surface, leading to reproducible detection of the antigen-Ab binding events. Adsorption of Abs on the gold electrode resulted in the most heterogeneous surface covering and the least repeatable measurements. Additionally, the immobilization layer configuration—on electrodes only (SAM layer, B), in between the electrodes only (APTES modification, C) or on both (PMMA layer, A)—impacted the overall performance of the sensing device. The binding events occurring in between the electrodes were playing a major role in the overall change in capacitance compared to the ones occurring directly at the surface of the electrodes [19]. Therefore, we believe that the SAM deposition of the capture molecule (via the use of a linker or not) on interdigitated electrodes should not be indicated as it leads to the deposition of the probe only at the top of the electrodes.

### 4.3. Impact of Surface Cleanliness and Contamination

Significant variations in reported results among different studies may arise from the lack of homogeneity and reproducibility of the coating procedure, as a result of improper electrode cleaning before functionalization [4,94]. In the context of mercapto-alkyl SAM formation on gold electrodes, Love et al. discussed the importance of proper electrode cleaning procedures to achieve uniform coatings [18]. SAM formation is based on exchange process, suggesting that thiolated molecules can displace miscellaneous contaminants adsorbed at the electrode surface. However, the presence of contaminants greatly affects the kinetics of the reaction, and therefore its reproducibility. To achieve reproducible coatings, the electrodes can be cleaned with piranha solutions or oxygen plasma [18], or via electrochemical methods [14] in the case of metallic electrodes. In 2010, Bhalla et al. compared the efficacy of piranha, plasma, reductive and oxidative cleaning methods on micro-fabricated chips used for EIS detection [94]. By analyzing cyclic voltammetry scans, scanning electron microscope images and capacitance measurements, the authors demonstrated that the two electrochemical cleaning techniques could effectively remove contaminants from the chips without degradation. However, the reductive pathway may lead to the deposition of materials on the conducting surface. Therefore, oxidative electrochemical treatment was found to be the most suitable and reproducible method for cleaning gold electrodes [94]. 

### 4.4. Non-Specific Adsorption

Non-specific adsorption of molecules has a critical impact on biosensing measurements, especially when using capacitive detection [10,62]. Any molecule immobilized at the surface of a capacitive sensor through non-specific interactions may result in false-positive detection, and therefore greatly reduces its selectivity. Such matrix effect was described by Liao and Cui, in the context of capacitive detection of platelet-derived growth factor. The study demonstrated the beneficial effect of electrode potential sweeping in potentiostatic EIS for discriminating between specific target binding and non-specific adsorption of biomolecules at the surface of the sensors [65]. Although, despite this optimization, the ratio of the positive to negative control was still around 10:1. By increasing the background noise, the matrix effect can also decrease the sensitivity of the studied device. For example, the detection of Herpes virus 1 reached a LoD value of 0.21 fM in neat serum while the attomolar detection range was achieved in pure buffer [33]. 

A variety of anti-biofouling strategies—classified as active or passive techniques—were explored for many biomedical applications, such as bioelectronic devices, biosensors, nanoparticles, dental implants or polymeric materials [95,96,97,98,99]. Physical and chemical passive methods include the addition of adsorption blocking agents and the addition of a repelling chemical layer based on a polyethylene glycol (PEG) layer, alkanethiol SAM layer, or zwitterionic polymer. Controlling the extent of biomolecule adsorption may also be achieved by changing the surface topography. Active methods, on the other side, create shear forces that are stronger than the forces causing non-specific adsorption. They can be generated through acoustic waves generation, pressure-driven flow, or from electrical or mechanical transducers [95]. To our knowledge, only passive methods were reported to reduce non-specific adsorption of biomolecules on capacitive sensing surfaces. 

Among physical methods, the addition of bovine serum albumin (BSA) as a blocking agent was reported for the detection of enzymes [35], Abs [34], disease protein markers [8,14,67], viruses [26,32] or cells [73]. The use of biotin [74] or glycine [53] was shown to minimize non-specific binding events. Additionally, the addition of concentrated solutions of KCl was found to greatly reduce non-specific adsorption by disrupting weak interactions. Dijskma et al. showed that the injection of 100 mM KCl solution completely remove interferon gamma from gold surface without damaging the SAM functionalized layer [75]. 

Among chemical methods, the addition of an anti-biofouling PEG layer was highlighted in DNA-hybridization and interleukin biosensors [57,76]. Miranda-Figueroa et al. demonstrated the beneficial effect of added PEG chains on malaria biosensors. Not only the matrix tolerance was improved, but also the LoD adynamic detection range were enhanced [30].

The design of suitable strategies against nonspecific binding highly depends on the nature of the targeted analytes, therefore requiring extensive trial iterations. Dykstra et al. developed a microfluidic platform that can measure protein adsorption on selected surfaces. This device offers the possibility to rapidly screen various materials toward their tendency to repel biomolecules, and could be of great interest for the design of capacitive biosensors in the future [100].

### 4.5. Surface Insulation and Coverage 

The surface of capacitive sensors must be insulated and hole-free to avoid charges to move through the layer, leading to the apparition of a faradaic current between the conductors [3,19,20,71], that would result in a change of capacitance of the surface and therefore a decrease in sensitivity [10]. Common insulating strategies relies on SAM covering based on alkyl-thiols, polymeric layers and silanization [19].

In the context of gold-thiol SAM formation, alkylthiols are added to insulate the sensing layer. Mirsky et al. reported that long chains should be privileged as short chains were prone to desorption. Proper insulation of gold electrodes was achieved with 15-/16-mercaptohexadecanoic acid [73]. Later, dodecanethiol [21,39,40,76,77,78,88], hexadecanethiol [101], mercaptohexanol [102], and mercapto-undecanol [4] were extensively used to insulate gold electrodes.

The quality of surface insulation largely depends on the selection of the functionalization procedure. Rickert et al. studied the insulation of epitope-modified gold electrodes with hydroxyundecanethiol (HUT). Simultaneous adsorption of a mixture of HUT and peptide was compared to the sequential adsorption of both components. The adsorption of mixed solutions resulted in poorly reproducible functionalization. On the contrary, reproducible and highly resistive films were obtained when the HUT was adsorbed after the epitope was immobilized [28].

In addition to provide chemical functionalities for the post-conjugation of capture biomolecules, polymeric layers were reported for the insulation of conductive electrodes. The insulation of Abs-modified gold electrodes with a 50 nm polytyramine film led to the detection of HSA down to 1.6 ng/mL concentration and with high reproducibility [72]. The quality of the insulating layer was probed by cyclic voltammetry. Berney et al. developed a capacitive detector for transferrin and studied the effect of PEG, as a non-conductive polymer, to insulate the sensing surface. When transferrin Ab was immobilized on non-insulated surface, the capacitance measurements after exposition to the targeted antigen were not reproducible. The addition of a PEG overlay system indicated the possibility to develop differential capacitive biosensors. However, the lack of continuity and integrity of the PEG layer did not allow for quantitative measurement of transferrin [20].

In conclusion, several requirements must be followed when designing and preparing a sensing surface for capacitive biosensors. First, the surface must be free of contaminants and prepared in as clean conditions as possible. Then, an insulation layer must be present to avoid faradaic currents that would lead to a drastic decrease in sensitivity. In the case of deposition of an oxide or polymeric layer on top of the electrodes, this layer should be however as thin as possible to keep good sensitivity properties. Finally, non-specific adsorption should be avoided to reduce false-positive results. Toward this goal, the addition of BSA or PEG layers have been the most reported technique.

## 5. Conclusions and Perspectives

Capacitive biosensors could greatly contribute to various diagnostic applications. Compared to other classic sensors, they offer the opportunity to develop in rapid and non-expensive portable sensors. Since their first development 30 years ago, capacitive biosensors were mainly applied to virus detection, and cancer or inflammatory diseases diagnostics. Several limitations restrained their expansion to other clinical applications. In particular, non-faradaic impedimetric sensors generally suffered from poor measurements reproducibility [4,12,13,19] and large standard deviations, as a result of non-optimal surface characteristics. Among the parameters which were evaluated to improve the performance of capacitive biosensors, one might concentrate on the following aspects: (i) surface cleaning by electrochemical treatment proved to increase the repeatability of subsequent functionalization steps; (ii) selection of the immobilization strategy in combination with proper insulation techniques showed to have drastic impact on the resulting sensor sensitivity; and (iii) the addition of blocking agents to mitigate non-specific adsorption events resulted in enhanced specificity and sensitivity.

Considering these multiple parameters, we can expect that the next generation of capacitive biosensors will result from rational design of the surface composition, morphology and geometry to enlarge the scope of these sensors in clinical applications. 

## Figures and Tables

**Figure 1 biosensors-13-00017-f001:**
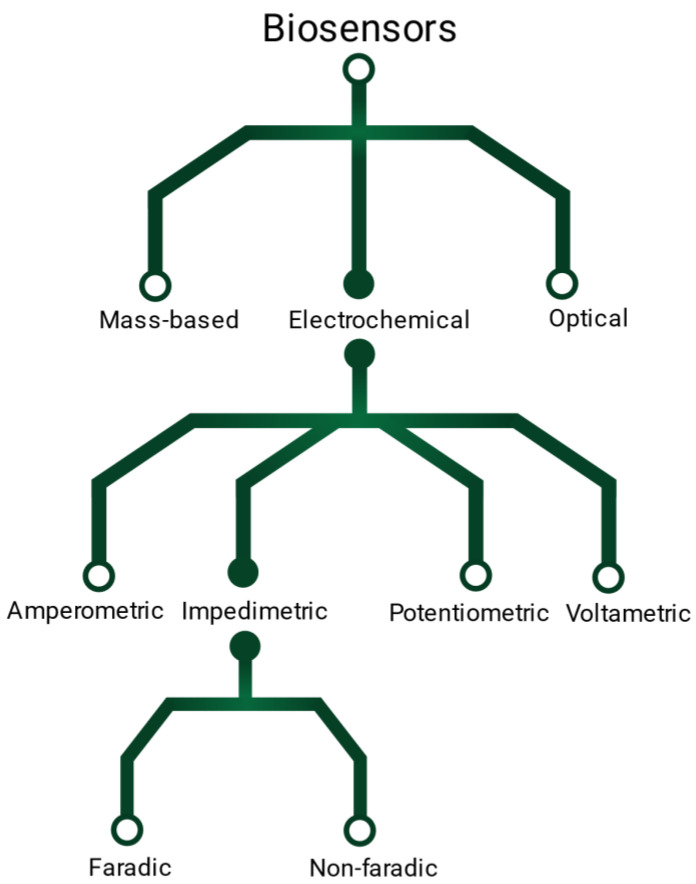
Illustration of the different types of electrical sensors.

**Figure 2 biosensors-13-00017-f002:**
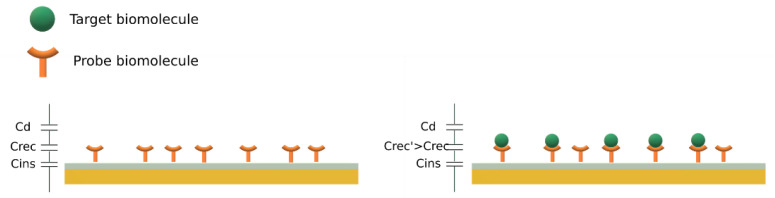
Schematic representation of electrode–solution interfaces for capacitive detection of biomolecules. C_rec_ increases after capture of the target biomolecules.

**Figure 3 biosensors-13-00017-f003:**
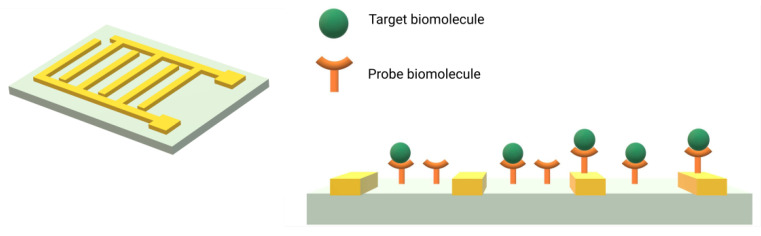
Graphical representation of interdigitated electrodes on a surface (**left**) and cross section view of an interdigitated biosensor after probe capture (**right**). Adapted from Tsouti et al. [3].

**Figure 4 biosensors-13-00017-f004:**
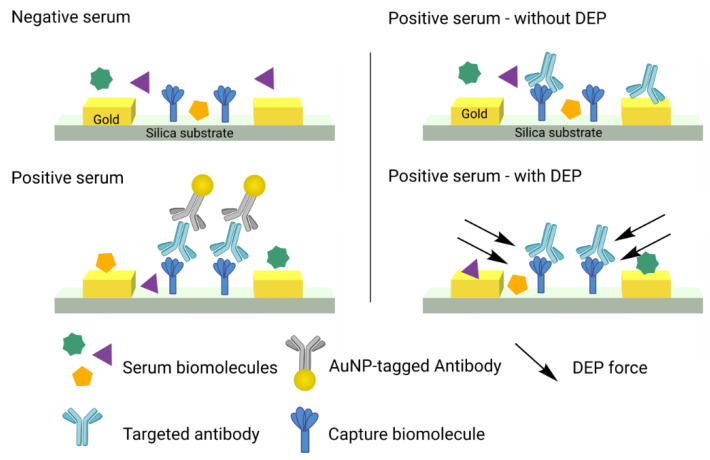
Enhancement of SARS-CoV-2 antibodies detection via the use of AuNPs (**left**) or dielectrophoresis force (**right**). Adapted from Zeng et al. 2022 [34].

**Figure 5 biosensors-13-00017-f005:**
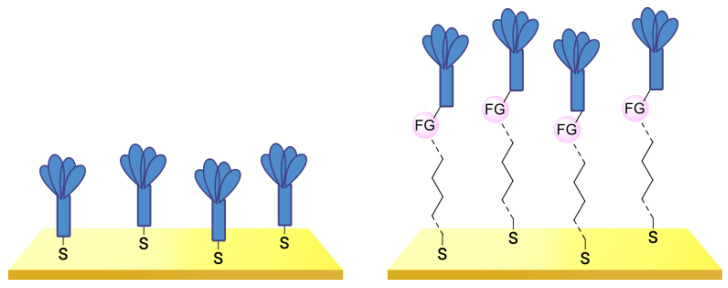
Comparison of direct SAM formation of thiol-containing biomolecules (**left**) and use of a thiol linker (**right**) for the immobilization of biomolecules at the surface of a metallic electrode.

**Figure 6 biosensors-13-00017-f006:**
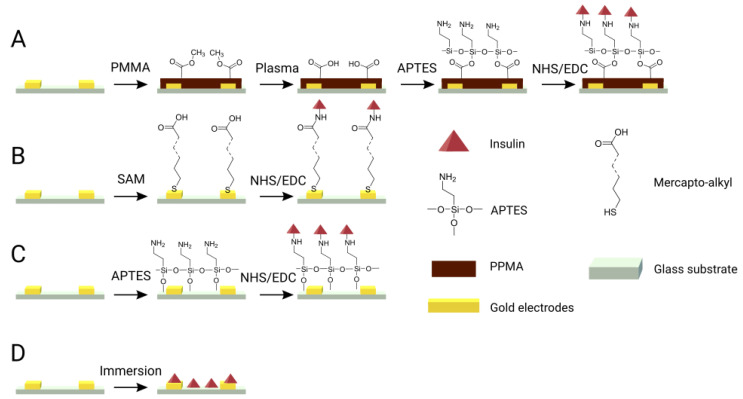
Schematic representation of the different immobilization strategies and architectures for the conjugation of insulin on IDEs. (**A**) spin-coated PMMA deposition; (**B**) mercapto-alkyl SAM deposition, (**C**) APTES covering; and (**D**) passive adsorption. Adapted from Castiello et al. [19].

**Table 1 biosensors-13-00017-t001:** Non-faradaic impedimetric biosensors reported for clinical applications. Capacitive sensors for food and waste control were not included.

	Target	Sensor Type	Electrode Material	Sensor Preparation	Detection Range	LoD	Ref.
Viral infection	Foot and Mouth disease	Immunosensor	Gold	SAM formation of thiol-modified epitope	N/D	N/D	[28]
	Hepatitis B	Immunosensor	Gold nanoislands	Parylene coating, followed by glutaraldehyde cross-linking	0.1–1000 ng/mL	<100 pg/mL in both buffer and serum	[29]
	Influenza H5N2	Immunosensor	Gold	Magnetic nanobeads coated with antibodies	1.5·10^1^–1.5.10^5^ ELD50/ml	1.6·10^2^ ELD_50_/mL of purified virus	[27]
	Influenza H5N1	Immunosensor	Gold	Antibody immobilization through adsorbed Protein A	10^1^–10^5^ EID50/mL	10^3^ EID_50_/mL in buffer	[26]
	Norovirus	Immunosensor	Gold	Polyaniline followed by streptavidin coupling and biotinylated Ab immobilization	1 fg/mL–1 ng/mL	60 ag/mL in buffer	[30]
	Zika virus	Immunosensor	Polyvinyl alcohol, Alignate and Polyaniline	EDC/NHS coupling of antibodies on alginate	N/D	6.6 × 10^3^ PFU/mL in buffer	[31]
	SARS-CoV-2 virus	Immunosensor	Poly(3,4-ethylenedioxythiophene) polystyrene sulfonate	Antibodies adsorption	N/D	147 TCID_50_/mL of virus from culture fluid	[32]
		Genosensor	Platinium/Titanium	APTES modification, followed by phosphoramidite linkage	5 µM–10 nM	10 nM in PBS	[60]
	SARS-CoV-2 antibodies	Immunosensor	Gold	APTES modification, followed by EDC/NHS coupling of antibodies	N/D	200 ng/mL and 2 μg/mL in buffer (for AuNP and DEP enhancements, respectively)	[34]
	Cytomegalovirus	Genosensor	Gold	SAM formation of thiolated oligonucleotides	N/D	0.2 aM of pure fragment in buffer	[13]
	West Nile virus	Genosensor	Gold	SAM formation of thiolated oligonucleotides	3–33 aM	1.5 aM in buffer	[4]
	Herpes 1 virus	Genosensor	Aluminum	SAM formation of thiolated oligonucleotides	0.2 fM–0.2 pM in serum	10.7 aM in buffer 0.21 fM in neat serum	[33]
	Papillomavirus	Genosensor	Indium oxide	SAM formation of thiolated oligonucleotides	0.1 pM–0.1 μM	20 fM in buffer	[61]
Pathogen markers	Malaria enzyme marker	Aptasensor	Graphene oxide	Graphene modified aptasensors linked to grapehene surface via π- π stacking	0.78 fM–100 nM	0.78 fM in diluted human serum	[35]
		Aptasensor	Gold	Thiolated-aptamers Polyethylene glycol layer added to reduce non-specific adsorption	1 pM–100 nM	0.77 fM in 50% human serum 1.49 pM in whole human serum	[62]
	*Schistosoma japonicum* antigen	Immunosensor	Gold	Antibody immobilization through adsorbed Protein A	0.4–18 ng/mL	0.1 ng/mL in PBS	[39]
Cancer biomarkers	SSAT	Immunosensor	Gold/Titanium	Parylene coating, followed by glutaraldehyde cross-linking	1.25–10 mg/L	1.25 mg/mL in buffer	[9]
	Her2	Aptasensor	Gold	SAM formation of thiol-modified aptamers	1 pM–100 nM	0.1 ng/mL in non-diluted serum	[46]
		Aptasensor	Gold	SAM formation of mercaptopropionic acid followed by peptide coupling	0.2–2 ng/mL	0.2 ng/mL in diluted serum	[8]
	Her4	Affimer-based sensor	Gold	SAM formation of cysteine-modified aptamers	1 pM–100 nM	1 pM in non-diluted serum	[63]
	PMSA	Protein-affinity-based sensor	Aluminum electrodes	Carboxylic-modified gold nanoparticles layer formed via thiol-gold bond, followed by peptide coupling	10 pM–100 nM	10 pM pure antigens in buffer	[64]
	Platelet derived growth factor	Aptasensor	Silicium	APTES modification followed by phosphoramidite bond	1–50 µg/mL	1 µg/mL in buffer	[65]
	Squamous carcinoma antigen	Immunosensor	Gold	Carboxylic acids introduced via SAM formation, followed by peptide coupling	N/D	2.43 μg/mL in buffer	[14]
Chronic or inflammatory diseases	Protein C reactive	Immunosensor	Gold	Carboxylic acids introduced via SAM formation, followed by peptide coupling	25 pg/mL–25 ng/ml	25 pg/mL in PBS	[11]
		Aptasensor	Gold	SAM formation of thiol-modified aptamers	200 pg/mL–2 ng/mL	200 pg/mL in PBS	[12]
		Immunosensor	Gold	ZnO thin film deposition, followed by succinimidyl propionate crosslinking	0.01–20 µg/mL.	0.10 µg/mL in human serum and whole blood	[47]
		Immunosensor	Gold	Carbon fibers sputtered on the electrode, followed by dithiobis(succinimidyl) propionate cross-linking	1 fg/mL–1 ng/mL	10 fg/mL in PBS and serum buffer	[48]
		Immunosensor	Nickel	SAM formation of carboxylic acids, followed by	1–250 ng/ml	1 ng/mL of purified antigens	[66]
	Myeloperoxidase	Immunosensor	Gold	Immobilization of streptavidin via SAM formation, followed by biotinylated-antibodies conjugation	1 pg/mL to 1 μg/ml	~1 pg/mL in buffer	[49]
	Troponin	Immunosensor	Screen printed electrode	Gold nanoparticles spread on the electrode, followed by adsorption of the antibody	0.1–12.5	0.2 ng/mL in buffer	[67]
		Immunosensor	Alumiium	Amino groups introduced via SAM formation, followed by cross-linking with glutaraldehyde	0.01–5 ng/mL in PBS buffer 0.07–6.83 ng/mL in human blood serum	0.01 ng/mL in PBS 0.07 ng/mL in human serum	[68]
	Human chitinase-3-like protein 1	Immunosensor	Gold	Thiourea introduction via SAM formation, followed by glutaraldehyde cross-liking	0.1 μg/L–1 mg/L	0.07 μg/L in buffer	[50]
	Transferrin	Immunosensor	Silicon doped semiconductor	Introduction of amine followed by glutaraldehyde cross-linking	N/D	N/D	[20]
		Immunosensor	Glass carbon	Electropolymerization of phenylenediamine, followed by glutaraldehyde cross-coupling	0.1–45.0 ng/mL in	0.061 ng/mL in PBS	[52]
	Interleukin-3	Immunosensor	Zeolite-Iron	Amine introduction followed by peptide coupling	3–100 pg/mL	3 pg/mL in buffer	[51]
	Interleukin-6	Immunosensor	Gold	Carboxylic acids introduced via SAM formation, followed by peptide coupling	25 pg/mL–25 ng/ml	25 pg/mL in PBS	[11]
	Nampt	Aptasensor	Gold	SAM thiol aptamers	1–50 ng/mL	1 ng/mL in diluted serum	[53]
	Amyloid beta	Aptasensor	Platinium/Titanium	Amine introduction followed by peptide coupling	0.001–10 μM	1 fg/mL in buffer	[54]
	LDL-Cholesterol	Immunosensor	Gold	Amino groups introduced via SAM formation, followed by cross-linking with glutaraldehyde	N/D	120 pg/mL of pure antigens	[6]
Unicellular organisms	Peripheral blood mononuclear cell	Immunosensor	Gold	Carboxylic groups introduced via SAM formation followed by peptide coupling	N/D	10^4^ cells/mL in buffer	[34]
	CD34^+^ T-cells	Immunosensor	Gold	Carboxylic acids introduced via SAM formation, followed by peptide coupling	50–1 × 10^5^ cells/mL	44 cells/mL in diluted serum	[69]
	Bacteria answers to antibiotics	/	Tantalum silicide	Polyethyleneimine layer for bacteria adsorption (non-specific)	N/D	N/D	[57]

Abbreviations: N/D: not detailed. TCID_50_: tissue culture infective dose. ELD_50_: 50% Egg Lethal Dose. EID_50_: 50% Egg Infective Dose. APTES: aminopropyltriethoxy silane. SAM: Self-assembled monolayer.

**Table 2 biosensors-13-00017-t002:** Methods reported for the covalent conjugation of capture biomolecules on insulating layers. R: Capture molecule.

Function Introduced	Coupling Agent	Capture Molecule	Chemical Linkage	Reference
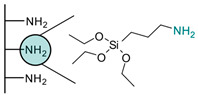 3-Triethoxysilylpropylamine	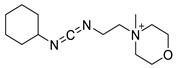 N-Cyclohexyl-N′-(2-morpholinoethyl)carbodiimide	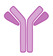 Antibody	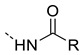 Peptide bond	[86]
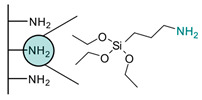 3-Triethoxysilylpropylamine	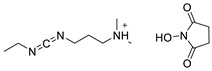 EDC/NHS	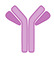 Antibody	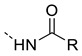 Peptide bond	[19]
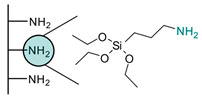 3-Triethoxysilylpropylamine	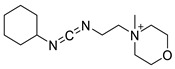 N-Cyclohexyl-N′-(2-morpholinoethyl)carbodiimide	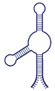 Aptamer	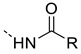 Peptide bond	[65]
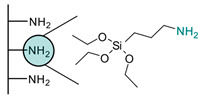 3-Triethoxysilylpropylamine	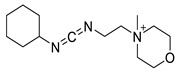 N-Cyclohexyl-N′-(2-morpholinoethyl)carbodiimide	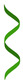 ssDNA	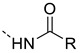 Peptide bond	[60]
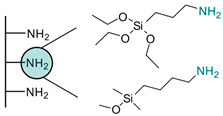 3-Triethoxysilylpropylamine,Aminobutyldimethylmethoxysilane	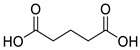 Glutaraldehyde	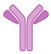 Antibody	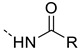 Peptide bond	[20]
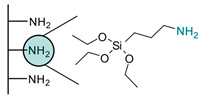 3-Triethoxysilylpropylamine	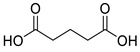 Glutaraldehyde	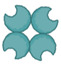 Avidin	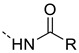 Peptide bond	[82]
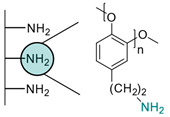 Polytyramine	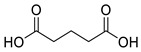 Glutaraldehyde	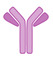 Antibody	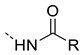 Peptide bond	[70]
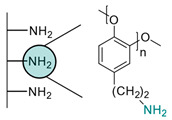 Polytyramine	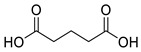 Glutaraldehyde	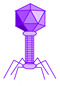 Phage	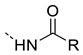 Peptide bond	[76]
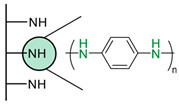 Polyaniline	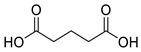 Glutaraldehyde	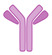 Antibody	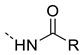 Peptide bond	[85]
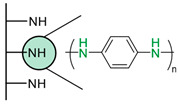 Polyaniline	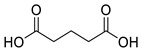 Glutaraldehyde	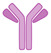 Antibody	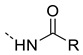 Peptide bond	[83]
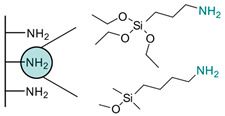 3-Triethoxysilylpropylamine,Aminobutyldimethylmethoxysilane	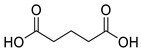 Glutaraldehyde	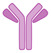 Antibody	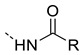 Peptide bond	[81]
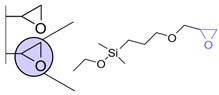 Glycidoxypropyldimethylethoxysilane	None	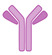 Antibody	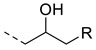 Epoxide ring-opening
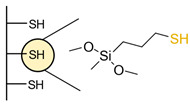 Mercaptopropylmethyldimethoxysilane	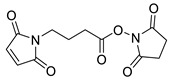 N-y-maleimidobutyryloxy succinimide ester	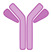 Antibody	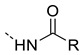 Peptide bond	[55]
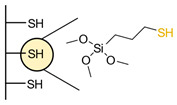 Mercaptopropyltrimethoxysilane	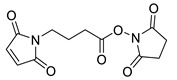 N-y-maleimidobutyloxy succinimide ester	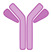 Antibody	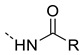 Peptide bond	[41]
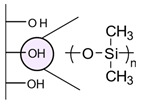 Polydimethylsiloxane	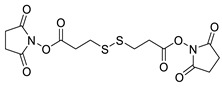 Dithiobis (succinimidyl propionate)	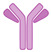 Antibody	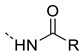 Peptide bond	[47]

## Data Availability

We do not present original data in the manuscript as it is a review article.

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
