# Peer review of "Design and Preparation of Sensing Surfaces for Capacitive Biodetection"

_biosensors, 2022, doi:10.3390/bios13010017_

Round 1

Reviewer 1 Report

The review entitled "Design and preparation of sensing surfaces for capacitive biodetection" is a comprenhensive review about capacitive detection and the methods for surface preparation. The paper is well written, and organized and it covers the most relevant state-of-art sensors that use in a way or another the preparation of sensing surfaces. 

But eventhough all the state of art is well covered I think the authors should cover a bit more in-depth the background of each bonding method.

For instance, in 4.2.1 it is not clear the difference between thiol and alky-thiol methods. Maybe the authors could deep into the chemical linkage methods and the differences between them and maybe a summary of when one is more convenient than the other.

In section 4.2.2 covalent immobilization a comparison of this method with the previous one is missing when is this more suitable than the previous one would be nice.

Moreover, there is no discussion between the two proposed methods Syanization and polymerization. The type of chemical linkages are not explained in detail either.

Also, it would be good to provide a more in-depth description of which types of molecules require cross-linkers, and which ones do not.  And when to use carbamide or phospahmides.

There are two minor typos:

 line 49 that a comma is misplaced

line 412 tin should be thin.

Reviewer 2 Report

This review focuses on the potential causes of poor reproducibility of capacitive biosensors, an issue that is important in sensor development. The manuscript is informative, systematic, and worthy of publication.

However, during my reading, I felt that Section 3 was a bit lengthy, of poor importance to the issue, and has already been covered in previous review articles, and I would advise that the section be adequately condensed.
